# Pollution Assessment Based on Element Concentration of Tree Leaves and Topsoil in Ayutthaya Province, Thailand

**DOI:** 10.3390/ijerph17145165

**Published:** 2020-07-17

**Authors:** Vanda Éva Molnár, Edina Simon, Sarawut Ninsawat, Béla Tóthmérész, Szilárd Szabó

**Affiliations:** 1Department of Physical Geography and Geoinformatics, University of Debrecen, H-4032 Debrecen, Hungary; molnarvandaeva@gmail.com (V.É.M.); szabo.szilard@science.unideb.hu (S.S.); 2Department of Ecology, University of Debrecen, H-4032 Debrecen, Hungary; 3Remote Sensing and Geographic Information Systems (RS&GIS) FoS, Asian Institute of 5 Technology (AIT), Klong Luang, Pathumthani 12120, Thailand; sarawutn@ait.ac.th; 4MTA-DE Biodiversity and Ecosystem Services Research Group, H-4032 Debrecen, Hungary; tothmerb@gmail.com

**Keywords:** urbanization, metals, plants, air pollution, soil pollution, soil humus

## Abstract

Atmospheric aerosol particles containing heavy metal contaminants deposit on the surface of plant leaves and the topsoil. Our aim was to reveal the pollution along an industrial–urban–rural gradient (IURG) in the central provinces of Thailand. Leaf samples from *Ficus religiosa* and *Mimusops elengi* were collected along with topsoil samples under the selected trees. Al, Ba, Ca, Cr, Cu, Fe, K, Mg, Mn, Na, Ni, Pb, and Zn concentrations were determined by ICP-OES in soil and plant samples. Soils were not polluted according to the critical value; furthermore, the elemental composition did not differ among the sampling sites of the IURG. The rural site was also polluted due to heavy amounts of untreated wastewater of the adjacent Chao Phraya River. Bioaccumulation factors of Ba, Cu, and Mn was higher than 1, suggesting active accumulation of these elements in plant tissue. Our findings proved that the deposition of air pollutants and the resistance to air pollutants in the case of plant leaves were different and that humus materials of the soils had relevant role in bioaccumulation of Al, Ba, and Cu. At the same time, the geochemical background, the source of pollution, and the local plant species greatly influence the metal content of any given environmental compartment.

## 1. Introduction

The proportion of urban population has been rapidly growing worldwide, and it is projected to increase in the future [1,2]. Urban and industrial areas, as centers of human activities, are diverse sources of various pollutants [3]. The extensive economic growth and the intensified anthropogenic activities cause severe air pollution [4,5]. Vegetation and soil are primary sinks for air pollution; the heavy metals translocated via atmospheric deposition negatively impact ecosystems [6]. The spillover effect between neighboring cities through certain linkages also increases the negative impact of air pollution [7].

Particulate matter (PM) is a typical pollutant in urban areas; it is usually rich in toxic organic components and heavy metals [8,9]. Thus, PM pollution proposes serious health issues for the urban population [9]. As a result of human activities, the natural concentrations of heavy metals increase, and they can accumulate in the soil and the vegetation due to their non-biodegradable nature [10]. Unfortunately, atmospheric heavy metal emission (the amount of emitted pollutants at the source) and immission (measured concentration of pollutants in the atmosphere at a given location) levels are rarely monitored. In recent studies, elevated concentrations of several metals have been found in the soil around car repair shops and carwashes [11] and even in the soil of suburban vegetable gardens [12,13].

Urban trees are used for indirect monitoring of PM and heavy metals in urban environments due to their distribution and cost-effectiveness [14]. The accumulation of air contaminants by plants is widely documented [15,16,17,18,19]. Trees are particularly efficient in trapping; thus, reducing airborne particles that can deposit in tree leaves’ stomatal openings and waxy cuticles [14]. Leaves accumulate heavy metals in high quantities [20]. Naturally, there is also a significant uptake via the root system, but the intensity of translocation through plant parts is dependent on the given metal and species. According to Feng et al. [6], heavy metals that deposit from the atmosphere have greater bioavailability than that in soil based on rice consumption.

Although several authors studied the atmospheric deposition regarding both the quantities and elemental composition of dust [21,22,23,24,25], the comparisons among cities are not widespread, and finding the differences in a multivariate approach is rare. It is vital to reveal how the geochemical background, the source of pollution, and the investigated plant species can influence the distribution and cross-connections of heavy metals in different environments.

Soil organic matter (SOM) influences many soil parameters such as availability of nutrients and water management positively, and it is primarily responsible for soil fertility [26]. SOM has direct effect on the mobility of heavy metals and can impede or facilitate their bioavailability depending on the level of organic matter humification (from fulvic acids to humin substances) and the formed chelates [27,28]; thus, it generally plays an important role in metal retention [29]. However, both SOM content and humus quality are disturbed by human activities in urban soils, and as the level of disturbance changes in the urban environment, the role of natural and anthropogenic factors can vary. For example, Horváth et al. [30] found radial increase in soil organic matter content from the city center in Sopron city, Hungary. On the contrary, Oktaba et al. [31] observed that soils from the center of Pruszkóv Town in Poland contained more humic substances than those from the outskirts.

In this work, urban, industrial and rural sites were selected in the central provinces of Thailand. The aim was to assess the variability of heavy metal content in tree leaves and topsoil at the selected sites. The rapid urbanization and industrialization in Thailand, particularly in the Bangkok Metropolitan Region, has led to an increase in emissions [32]. The high density of motor vehicles has a large contribution to the local air pollution, which is amplified by the lack of proper urban planning [33]. We aimed to determine the level of pollution in industrial, urban, and rural areas and to reveal the potential of tree leaves as indicator of the environmental state. Thus, our hypotheses are the following: (i) There is higher heavy metal concentration in leaves samples from urban and industrial sites than from the rural; (ii) there is also higher heavy metal concentration in soil from urban and industrial sites than from rural, and (iii) tree leaves can reflect the heavy metal concentration of the environment.

## 2. Materials and Methods

### 2.1. Study Sites and Sample Collection

There were three sampling areas (industrial, urban, and rural) in Thailand (Figure 1). The industrial area is in Wat Klong Phut-Sa near the Bangpa-in Industrial Estate in Phra Nakhon Si Ayutthaya Province. There are 109 factories in the Bangpa-in Industrial Estate area which is divided into five areas: Industrial Area, Free zone, Commercial and Residential area, Utility area and facilities, and Green area. The urban area was found at the Rangsit campus of Thammasat University, which was in Khlong Luang, Pathum Thani, 42 km North of Bangkok. The campus is divided into three areas: academic zone, housing zone (dormitories), and various sport facilities. The rural area was in Wat Pho Teang Nuea in Pho Taeng, Bang Sai, Phra Nakhon Si Ayutthaya. It is nearby the Chao Phraya River. Sacred fig (*Ficus religiosa*) and indian medlar (*Mimusops elengi*) were chosen because these species were found in all three sampling areas (industrial, urban, and rural). Three individual trees were chosen randomly from each species, and 15 leaves were collected from each tree at 1.5 m high. Thus, we collected 90 leaves per sampling sites. At each sampling site, six soil samples were collected at a depth of 0–20 cm along the urbanization gradient from each sampling area from the same micro-location near the selected trees.

### 2.2. Elemental Analysis in Leaves and Soil Samples

After sample collection, leaves were washed with tap waters [34]. Because using distilled water may cause remarkable differences in the elemental concentration of leaves by the osmotic effects, tap water was used for washing. Leaf samples were dried for 24 h at 60 °C; then, the samples were homogenized and stored in plastic tubes until pre-treatment. Soil samples were dried for 24 h at 105 °C. From each soil sample, 50–100 g soil was dried at room temperature. After drying, stones, plant roots, and residues were removed with plastic tweezers. Samples were sieved in 2-mm plastic sieve. Then the samples were homogenized with agate mortar and stored in plastic tubes until pre-treatment. For elemental analysis, 0.2g of plant tissue and 0.2 g soil sample was digested using 5 mL 65% (m/m) nitric acid and 1 mL 30% (m/m) hydrogen-peroxide in a microwave digestion unit (Milestone 1200 Mega) for 5min in 300W and subsequently 5min in 600W. Digested samples were diluted to 25 mL with deionized water.

Inductively coupled plasma optical emission spectrometry (ICP-OES 5110 Agilent Technologies) was used during the elemental analysis of leaf and soil samples. We used six-point calibration procedures with multi-element calibration solution (Merck ICP multi-element standard solution IV) and measured the concentration of Al, Ba, Ca, Cr, Cu, Fe, K, Mg, Mn, Na, Ni, Pb, and Zn.

### 2.3. Determination of Soil Organic Matter Quantity and Humus Quality

Organic matter content was determined titrimetrically by the Tjurin method [35]. One hundred mg of the soil samples was weighted into an Erlenmeyer flask; after adding 10 mL of the oxidizing solution (0.4 N K_2_Cr_2_O_7_:H_2_SO_4_—1:1), the samples were boiled on a hot plate for 3 min. The cooled mixture was diluted to 100 mL with distilled water, and then, the residual dichromate was determined by titration with 0.2 N Mohr’s salt (Fe(NH_4_)_2_(S0_4_)_2_^.^6H_2_0) using ferroin as indicator. Soil humus quality was determined by the Hargitai method [36]. Dried soil (10 g) was measured twice from each sample into Erlenmeyer flasks; then, one of the flasks was filled with 0.5% NaOH up to the 100 mL mark, while the other one was filled with 1% NaF. After thoroughly shaken, the flasks were left for 48 h. During this time, NaOH dissolves lower quality humus substances (fulvic acids), while NaF dissolves good quality, real humus substances (himatomelan acids, grey and brown humic acids) [37]. The extinction of the solutions was measured at 533 nm on a Uvi Light XS2 spectrophotometer (SCHOTT Instruments). Humus stability coefficient (K) was calculated as follows:
K = E_NaF_/(E_NaOH_*H),(1)
where E_NaF_ is the extinction value of the NaF extract, E_NaOH_ is the extinction value of the NaOH extract, and H is the total humus content.

### 2.4. Calculation of Bioaccumulation Factor (BAF)

The level of bioaccumulation is characterized with the bioaccumulation factor (BAF) defined as a ratio of concentration of trace elements in the plant and the concentration of trace elements in the surrounding environment [38,39]:
BAF = C_org_/C_m_,(2)
where C_org_ means the concentration of trace elements in leaves and C_m_ is the concentration of trace elements in the soil. Measured heavy metal content of the soils with hydrogen-peroxide and microwave digestion shows almost a total concentration. Some elements cannot be extracted with digestion; thus, we refer to them as extractable with strong extractants.

### 2.5. Statistical Analysis

Statistical analysis was performed using the Statistica 7.0 statistical software package. Principal Component Analysis (PCA) was used to display the effect of tree species and urbanization on the elemental concentration of leaves and topsoil. The normal distribution was tested with the Shapiro–Wilk test. The homogeneity of variances was tested with the Levene’s test. The elemental concentration of leaves and soil parameters were compared based on the studied areas and tree species by Generalized Linear Model (GLZ) [40].

We analyzed the relationship of plant species (as dummy variables, *Ficus religiosa* was coded as 0, *Mimusops elengi* was coded as 1), humus content and humus quality (as independent variables), and the BAF values (as dependent variables). We applied two models, the multiple linear regression (MLR) and the multivariate adaptive regression splines (MARS) regression. While the MLR has the prerequisite of linearity and the normal distribution of the model residuals, MARS is a non-parametric extension of MLR without assumptions on dependent and independent variables. MARS algorithm runs linear regressions by partitioning the data into linear sections. MARS models are based on two steps. First, values of independent variables are partitioned into groups, and linear regression models are performed on all groups. Each group has a linear function with its own slopes and where the connections of lines are called the knots. All knots have two basis functions (BFs). In the second step, the algorithm uses these BFs as independent variables to provide a final model, removing the BFs that contribute the least [41]. Furthermore, we applied the 2-fold cross validation with 25 repetitions, i.e., we had split the dataset into a training set that was used to build the models and a testing set to assess the model efficiency. Twofold cross validation means that the dataset is split only in two parts (50% training and 50% test sets), and its combination with 25 repetitions resulted in 50 models [42]. We reported the pseudo-R^2^ values, which is the correlation between the modelled and observed values of the data of the testing set. R^2^ values reflect the models’ efficiency and not the model fit. Statistical modelling was conducted in R 3.6.3.

## 3. Results and Discussion

### 3.1. Elemental Concentration in Leaves

The PCA biplot was used to display the elemental concentrations of leaves. The first component (PC1) contributed 48.6% of total variance and the second one (PC2) contributed 18.1%. Element concentration in leaves had significant difference (*p* < 0.05) by plants species along the PC1 axis, while the sites of urban gradient had not significant differences, i.e., there was no spatial pattern of the elements (Figure 2).

There was no significant difference in the elemental concentration of *M. elengi* leaves among the sites. Similar results were found for *F. religiosa*. The leaves of *M. elengi* contained higher amounts of Al, Cr, Fe, Na, and Pb, while the leaves of *F. religiosa* contained more Ca, Cu, K, Mg, and Zn (Table 1 and Appendix A).

The elemental concentrations in the leaf tissue of the tree species were found to be homogenous across the study sites (Appendix A); only in the case of Al, Ba, Cr, and Fe, there were significant differences (*p* < 0.05) according to GLZ. In the cases of other elements, the differences were insignificant. Usually, rural sites have the smallest metal loads due to smaller traffic and population density. However, in our case we note that the rural study site was alongside the Chao Phraya River, which is one of the major rivers of Thailand and is seriously polluted due to untreated wastewater originating from agricultural, industrial, and domestic activities [43]. Contaminants from the river can be transferred via natural processes into the soil of adjacent areas. This may explain that the metal content of plant tissue in the rural area was not different from that of the urban and industrial sites. Differences in Al and Fe can be explained with the geochemical background. Al is lightly soluble in the soil matrix and had the largest concentration in the urban sites but not in a toxic amount [44]. Fe is an essential element for plants, and its presence is related to clay minerals. Tree leaves of urban areas had the largest concentration but without threatening the normal plant growth [45]. However, Ba-concentration was the highest from leaves collected in the rural site, but in comparison with McBride et al. [46], where they studied non-polluted areas in urban gardens (New York City and Buffalo, USA), these values were 4–5 times smaller considering even the largest concentrations. The rest of the elements did not differ between the sampling sites, which can be the consequence of the low level of environmental loads by anthropogenic activities, and it is confirmed by the relatively low metal concentrations in the soils.

Nandy et al. [47] noted that *Ficus bengalensis, Alstonia scholaris,* and *Neolamarckia cadamba* species are tolerant to air pollution, while *F. religiosa* is sensitive to air pollution. Trees with thin leaves such as *F. religiosa* can receive less particles due to their leaf position. Despite this, trees with hard leaves can trap more air pollutants upwards or horizontally with their concave surface [48]. Prajapati and Tripathi [49] also demonstrated that *F. religiosa* can be used to intercept air pollutants. Based on the elemental concentration of leaves, we demonstrated that *M. elengi* is a more tolerant species than *F. religiosa. M. elengi* leaves include pericyclic fibers, rosette and prism shape calcium oxalate crystals, and resinous matter [50]. Changlian et al. [51] reported that *M. elengi* exhibited more resistance to air pollutants than others such as *Acmena accuminatissima*, *Lysidice rhodostegia*, *Bombax malabaricum,* and *Hibiscus rosa-sinensis*. Similar to earlier findings, our results indicated that the deposition of air pollutants and the resistance to air pollutants of plant leaves vary with structure, geometry, height, and canopy of the tree. Thakar and Mishra [52] demonstrated that smaller plants with short petioles and rough surface accumulate more air pollutants than larger plants with long petioles and smoother leaf surface.

### 3.2. Elemental Concentrations of Soil

There are numerous reports studying the heavy metal content of soils under anthropogenic influence. The first component (PC1) contributed 85.0% of total variance, and the second one (PC2) contributed 6.8%. There was no separation of sites based on soil elemental concentration (Figure 3).

Unlike earlier reports [53], our findings did not support that the concentrations of studied elements in the soil exceeded the standards of FAO [54] and WHO [55] in the studied Province of Thailand. Simasuwannarong et al. [56] also reported that the mean values of most heavy metals in the soil, except for As, were lower than Thailand’s soil quality standard for habitat and agriculture purposes and the worldwide background level. However, Krailertrattanachai et al. [57] demonstrated that road edge soils went from moderately to highly polluted with Cd, Cu, Pb, and Zn, and the concentration of these elements significantly decreased with increased distance from the roads. They also demonstrated that the Co, Cr, Cu, Ni, V, and Zn concentrations in some roadside agricultural soils were within or higher than the critical values of total trace metal concentrations in soils in Thailand that may be toxic to plants [57]. Sowana et al. [58] revealed that municipality areas, industrial zones, and dockyard areas in Pattani Bay, Thailand, had the highest potential for soil contamination by heavy metals, particularly Pb and As. Damrongsiri et al. [59] reported high heavy metal contamination (Cu, Pb, Zn, and Ni) in the topsoil in Thailand. We had not found any differences among the studied sites which could be the consequence of the proximity of Chao Phraya River at the rural site (Table 2). The river is seriously polluted due to untreated wastewater. At the same time, Mingkhwan and Worakhunpiset [60] reported that heavy metal levels in water did not exceed Thailand’s surface water quality standards, except for Mn in the studied Ayutthaya Province. Chaiyara et al. [61] also demonstrated that the concentrations of Cd, Cu, and Pb in the rivers were low compared to the marine standards of Chao Phraya River in Thailand and estuarine water elsewhere. The concentrations of Cd and Cu in sediments were similar to those in other rivers

### 3.3. BAF

The available proportion for plants can be lesser due to soil characteristics (i.e., amount of organic and inorganic colloids, pH, etc.) and the type of binding (adsorbed or within solid phase), and depending on the physiology of the given species, the accumulation varies even by the organs [62,63,64,65]. Thus, BAF can be a good indicator of accumulation, reflecting both availability of metals and the actual uptake in a given plant’s organ.

The BAF values were lower than 1 for all studied trace elements, except Cu and Mn (Figure 4). In the case of Cu, BAF values were higher than 1 in the industrial and rural sites based on the results of *F. religiosa.* While for Mn, higher BAF values than 1 were found in the industrial and urban sites based on *M. elengi* leaves. The BAF values of Cu and Mn were higher than 1 indicating that these elements could be accumulated in leaves from soil through any exposure route. Plants ingest heavy metals from 25 cm of the soil where the underlying foundations of most developed species are found [65]. Cu and Mn are essential micronutrients, but high densities of these metals have harmful impacts on plants and represent an ecological risk due to bioaccumulation [66,67,68]. Generally, the ability of plants to accumulate metals from the environment is highly dependent on the nature of the element, plant species, and all the cumulative or synergistic effects of biotic and abiotic factors.

As BAF is the indicator of the accumulation, and as humus content and the level of polymerization of the organic matter is the most important organic colloid of the soil, revealing their relationship is important. We proved the relevance of humus materials in case of Al, Ba, and Cu (R^2^ medians > 0.4), while in the case of Fe and Mn, the R^2^ medians were around 0.2, but the interquartile range indicated that the models can be more accurate in several cases. The relationship was the weakest in the case of Zn, with lowest upper quartiles. Results also pointed to the relevance of a robust approach, which came from the distribution: MARS was a better predictor algorithm than MLR. Splitting the database resulted in non-linear connections between the humus and the studied elements; thus, the MARS algorithm having no assumptions on distribution on the input data was more successful with better performance (Figure 5). Better performance of MARS had been proven in other studies too [69,70], and we also found it an effective prediction tool and, in this case, a technique for better exploration of relationships among different variables.

## 4. Conclusions

Distribution of heavy metals in the environment is relevantly affected by biotic and abiotic factors. We analyzed two species’ (*F. religiosa* and *M. elengi*) regarding the heavy metal content of their soil environment to reveal the pollution level. We determined that *F. religiosa* accumulated significantly higher metal content than *M. elengi* in the leaves in the cases of Ba, Ca, Cu, K, Mg, and Zn, while *M. elengi* had higher metal content in the cases of Al, Cr, Fe, Mn, Na, Ni, and Pb. According to the multivariate analysis, the species had a significantly different accumulation pattern. The location, the industrial–urban–rural gradient, had not significant difference regarding the metal uptake, i.e., the intensity of land-use had no discriminating effect which can be in connection with the polluted Chao Phraya River near the rural sampling site. Bioaccumulation (BAF) was significantly larger in *F. religiosa* especially in the industrial and rural sites. We revealed that the multiple linear regression was not as efficient in finding the relationship between metals SOM and the humus quality as the MARS regression which has no assumptions regarding linearity, distribution, and outliers. The applied k-fold cross-validation provided more reliable results related to a correlation because it uses a training and an independent testing dataset, and the repetitions provided 50 R^2^-values which showed the range of possible realizations instead of a single value. We proved the connection between bioaccumulation and humus in case of Al, Ba, and Cu. Our findings demonstrated that trees are a useful indicator of air pollutants for biomonitoring studies of cities.

## Figures and Tables

**Figure 1 ijerph-17-05165-f001:**
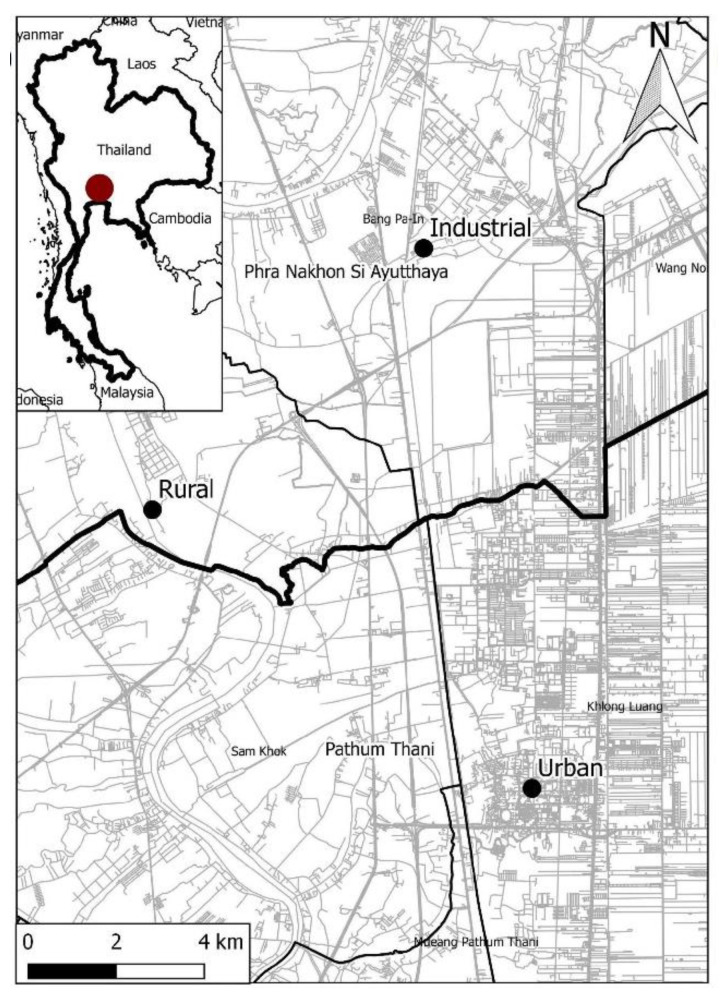
Sampling sites in Thailand. Notations: red point indicated the selected province in Thailand.

**Figure 2 ijerph-17-05165-f002:**
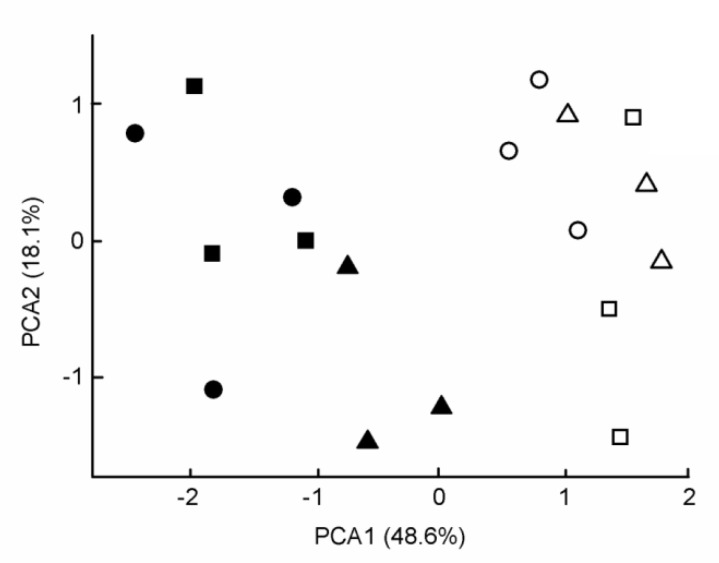
Principal component biplot of the elemental concentrations of leaves. Notations: 
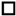
 = *Ficus religiosa* from industrial, 
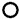
 = *F. religiosa* from urban, 
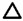
 = *F. religiosa* from rural, 
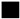
 = *Mimusops elengi* from industrial, 
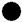
 = *M. elengi* from urban, 
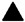
 = *M. elengi* from rural site.

**Figure 3 ijerph-17-05165-f003:**
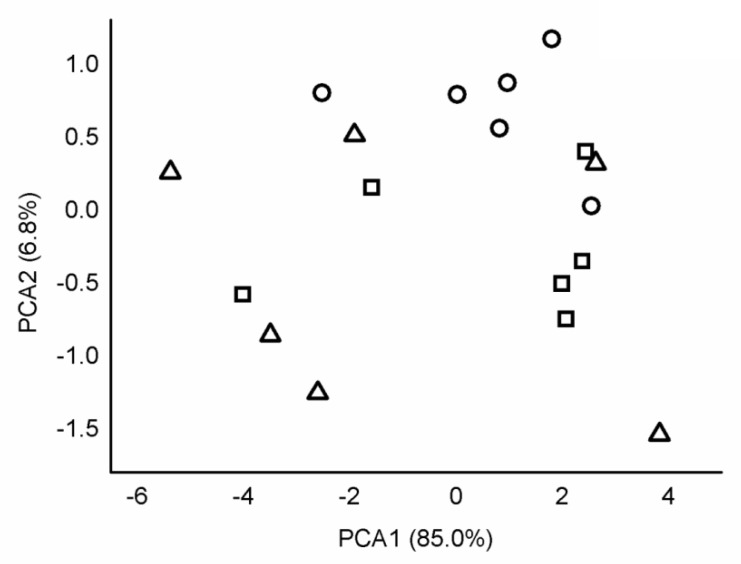
Principal component biplot of the elemental concentrations of soil. Notations: 
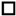
 = industrial site, 
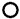
 = urban site, 
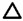
 = rural site.

**Figure 4 ijerph-17-05165-f004:**
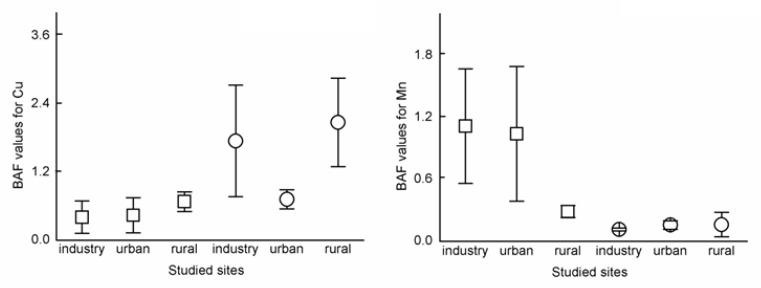
Values of Bioaccumulation factor (BAF) for Cu and Mn concentration of leaves and soil. Notations: 
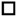
 = BAF value-based *M. elengi*, 
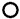
 = BAF value-based *F. religiosa.*

**Figure 5 ijerph-17-05165-f005:**
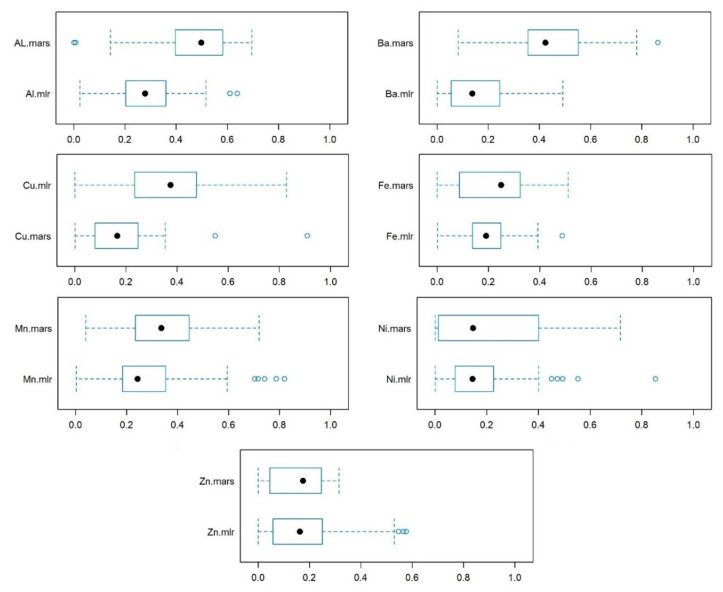
Relationship between humus content and elemental concentration of soil.

**Table 1 ijerph-17-05165-t001:** Mean ± standard error of elemental concentration (mean ± SE) of leaves in Thailand.

Elements	*M. elengi*			*F. religiosa*		
	Industrial	Urban	Rural	Industrial	Urban	Rural
Al, mg kg^−1^	160 ± 35	209 ± 53	123 ± 29	72 ± 16	124 ± 36	82 ± 25
Ba, mg kg^−1^	5.4 ± 0.8	5.4 ± 2.4	7.9 ± 1.4	7.3 ± 4.9	7.0 ± 0.8	17.9 ± 6.2
Ca, g kg^−1^	24 ± 3	20.0 ± 1.6	16.2 ± 1.0	37 ± 14	49 ± 5	36.0 ± 8.1
Cr, mg kg^−1^	1.6 ± 0.4	1.7 ± 0.4	0.4 ± 0.4	l.d.	0.8 ± 0.4	0.4 ± 0.4
Cu, mg kg^−1^	2.9 ± 0.4	2.5 ± 0.7	3.7 ± 0.4	13.9 ± 2.8	10.3 ± 0.3	13.2 ± 0.4
Fe, mg kg^−1^	167 ± 43	220 ± 55	139 ± 32	82 ± 8	125 ± 16	109 ± 19
K, g kg^−1^	8.7 ± 0.7	10.0 ± 2.3	12.1 ± 1.7	29 ± 2	30.7 ± 3.5	33.6 ± 2.3
Mg, g kg^−1^	2.2 ± 0.2	2.1 ± 0.3	2.3 ± 0.3	3.4 ± 0.7	5.5 ± 1.3	5.1 ± 0.6
Mn, mg kg^−1^	170 ± 81	168 ± 113	25 ± 6	23 ± 5	49 ± 23	24 ± 3
Na, mg kg^−1^	336 ± 84	533 ± 130	1314 ± 894	178 ± 72	185 ± 20	280 ± 63
Ni, mg kg^−1^	1.6 ± 0.4	1.6 ± 0.4	1.3 ± 0.1	1.2 ± 0.1	1.2 ± 0.1	1.2 ± 0.1
Pb, mg kg^−1^	1.2 ± 0.1	0.8 ± 0.4	0.4 ± 0.4	l.d.	0.4 ± 0.4	0.4 ± 0.4
Zn, mg kg^−1^	11.5 ± 2.0	9.5 ± 3.0	13.0 ± 2.3	23.6 ± 7.3	15.7 ± 3.9	26.0 ± 3.0

Notations: l.d. means limit of detection.

**Table 2 ijerph-17-05165-t002:** Mean ± standard error of soil parameters in Thailand.

Elements	Thailand			
	Industry	Urban	Rural	Critical Value
Al, g kg^−1^	36 ± 10	44 ± 7	22 ± 8	n.p.
Ba, mg kg^−1^	92 ± 17	83 ± 12	82 ± 33	n.p.
Ca, g kg^−1^	19 ± 6	6 ± 2	11 ± 8	n.p.
Cr, mg kg^−1^	33 ± 7	36 ± 12	21 ± 6	75–100
Cu, mg kg^−1^	16 ± 4	14 ± 3	18 ± 9	60–125
Fe, g kg^−1^	21 ± 4	27 ± 4	16 ± 4	n.p.
K, g kg^−1^	5 ± 1	5 ± 1	3 ± 1	n.p.
Mg, g kg−1	2 ± 0.5	2 ± 0.4	2 ± 1	n.p.
Mn, mg kg^−1^	207 ± 36	236 ± 67	225 ± 92	n.p.
Na, g kg^−1^	1 ± 0.2	1 ± 0.2	1 ± 0.3	n.p.
Ni, mg kg^−1^	13 ± 2	16 ± 3	11 ± 3	100
Pb, mg kg^−1^	11 ± 2	14 ± 2	13 ± 4	100–400
Zn, mg kg^−1^	40 ± 11	40 ± 7	87 ± 42	50–100
humus, %	2.5 ± 1.5	2.5 ± 0.6	3.4 ± 1.7	
stability coefficient of humus	1.5 ± 0.9	0.8 ± 0.2	0.9 ± 0.5	

The results of critical values are based on the study by Krailertrattanachai et al. (2019). N.p. means data was not published.

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
