# Peer review of "Pollution Assessment Based on Element Concentration of Tree Leaves and Topsoil in Ayutthaya Province, Thailand"

_ijerph, 2020, doi:10.3390/ijerph17145165_

Round 1

Reviewer 1 Report

This study mainly detected the concentrations of multiple elements in tree leaves and topsoils at selected industrial, urban, and rural sites of Thailand to explore the differences among areas, and between two tree species. The paper is with detailed method descriptions and good interpretation of results. There are some minor comments.

  1. Line 108, do you mean potassium dichromate “K2Cr2O7” instead of “K2Cr2O4”?
  2. Line 207~226 and Table 2, it would be better to list the levels of studied elements in other studies or the related standards, as well.

Author Response

Response to Reviewer 1 Comments

This study mainly detected the concentrations of multiple elements in tree leaves and topsoils at selected industrial, urban, and rural sites of Thailand to explore the differences among areas, and between two tree species. The paper is with detailed method descriptions and good interpretation of results. There are some minor comments.

Point 1: Line 108, do you mean potassium dichromate “K2Cr2O7” instead of “K2Cr2O4”?

Response 1: We corrected.

Point 2: Line 207~226 and Table 2, it would be better to list the levels of studied elements in other studies or the related standards, as well.

Response 2: Thank you so much your recommendation. We inserted the results of critical value to the Table.

Reviewer 2 Report

Dear Authors,

This paper spans the assessment of pollution (both essential plant elements and non-essential heavy metals) by measuring the total elemental concentrations in tree leaves and topsoil in Thailand.

The paper has some good background literature reviewed on the concerned problem of the studied area and with a good start and representation of results. However, there are some flow issues in the introduction section; the description of materials and methods were inadequate and weak discussion of obtained results (please see my specific comments on the PDF version of the manuscript). In most cases, there was a lack of consistency, coherency, and flow issues.

Also, this paper does not have any clearly stated objectives, and the study does not have any followed statistical design or sampling design. Moreover, the conclusion of the article did not draw from the obtained results of this study. In most cases, the conclusions were made based on the literature (very broad or vague sentences) instead of the findings/results obtained from the study.  

Based on the points mentioned above points, this paper needs a significant revision.

Best regards,

Author Response

Response to Reviewer 2 Comments

Dear Authors,

This paper spans the assessment of pollution (both essential plant elements and non-essential heavy metals) by measuring the total elemental concentrations in tree leaves and topsoil in Thailand.

The paper has some good background literature reviewed on the concerned problem of the studied area and with a good start and representation of results. However, there are some flow issues in the introduction section; the description of materials and methods were inadequate and weak discussion of obtained results (please see my specific comments on the PDF version of the manuscript). In most cases, there was a lack of consistency, coherency, and flow issues.

Also, this paper does not have any clearly stated objectives, and the study does not have any followed statistical design or sampling design. Moreover, the conclusion of the article did not draw from the obtained results of this study. In most cases, the conclusions were made based on the literature (very broad or vague sentences) instead of the findings/results obtained from the study.  

Based on the points mentioned above points, this paper needs a significant revision.

Response: We accepted and corrected all proposals indicated in the pdf which was uploaded into the server.

Reviewer 3 Report

This manuscript provides new knowledge on the effects of pollution on the elemental composition of tree leaves and topsoil in three different regions of a province in Thailand. There are a few comments that the authors need to consider for improvement of the manuscript.

.Table 1: There is no data from Hungary in this Table.

.For each plant species, ANOVA analysis should be carried out for the three different sites. The analysis should also be carried out for the data in Table 2.

.Include a discussion on the results of the ANOVA analysis.

.Page 8, lines 229-230: State which site is > 1 for Cu and Mn.

.Page 9: No height variation effect was studied and therefore the concluding remarks in lines 267-269 are not valid.

Author Response

Response to Reviewer 3 Comments

This manuscript provides new knowledge on the effects of pollution on the elemental composition of tree leaves and topsoil in three different regions of a province in Thailand. There are a few comments that the authors need to consider for improvement of the manuscript.

Point 1: Table 1: There is no data from Hungary in this Table.

Response 1: We corrected it was erratum.

Point 2: For each plant species, ANOVA analysis should be carried out for the three different sites. The analysis should also be carried out for the data in Table 2.

Response 2: ANOVA table (i.e. a GLZ was carried out, thus, we reported the related Chi2 and Log-likelihood statistics, and the p-values) can be found in the supplementary materials, in details.

Point 3: Include a discussion on the results of the ANOVA analysis.

Response 3: We inserted the results of the ANOVA analysis to the Results and Discussion section.

Point 4: Page 8, lines 229-230: State which site is > 1 for Cu and Mn.

Response 4: We inserted the statement with the details.

Point 5: Page 9: No height variation effect was studied and therefore the concluding remarks in lines 267-269 are not valid.

Response 5: We corrected.

Round 2

Reviewer 2 Report

Thanks for addressing the comments. 

Reviewer 3 Report

The authors have responded to my comments well. I recommend publication of this paper.